## [Peer Review File · Nature Communications]

Reviewers' comments:

Reviewer #1 (Remarks to the Author):

Majority influence in human social learning develops universally across seven societies
Edwin J. C. van Leeuwen, Emma Cohen, Emma Collier-Baker, Christian J. Rapold, Marie Schäfer,
Sebastian Schütte, Daniel B. M. Haun

This is an elegant and clever cross-cultural and developmental studies showing both variable and stable tendencies in children from 7 different societies aged between 4 and 14 years in their reliance on social information from peers in a 3 choice puzzle (3 possible tubes to drop one ball) in order to obtain a reward. With a sample of over 600 participants of various ages, not strictly matched in terms of age and numbers across 7 cultures that were, except for Germany (the reference, overloaded culture), opportunistically tested and with (as it stands) no other articulated rationale than being different and covering 4 continents.

Data are reported suggesting a) that across societies, children vary significantly in their overall tendency to rely on social information, and b) that across societies, there is a U shape developmental trend toward an early and late enhanced reliance on social information from a majority as opposed to a minority, with an invariant deflection or decrease of such tendency between 6 and 10 years of age.

These findings are certainly meaningful and stimulating, opening up new questions and demonstrating an economical way to study experimentally child development from a cross-cultural perspective, a long neglected domain. The paradigm is remarkably simple yielding rich information on the perennial issue of cultural factors in child development, in particular the variable and invariant aspects of the way children from different societies tend to rely on the larger (majority) group to learn and access information, the degree to which they are inclined to conform and learn from others in general, a traditional problem in both developmental and social psychology.

The apparently trans-cultural U shape developmental progression in the tendency to rely on majority as opposed to minority information is intriguing and begs for more research on the phenomenon. This research is needed to provide more specifics regarding the "changing of social motivations" proposed by the authors. The report leaves open the question of what is exactly the psychology of the younger children relying more on the majority rather than the minority information? How different is it from the older children showing the same trend? What accounts for the drop in-between? If the phenomenon holds, it could revive and would open a whole new body of cross-cultural and developmental studies.

This said, the weakest point of the paper -as I see it- is the acknowledged opportunistic choice of societies (except for Germany), with no other rationale provided nor any information regarding who the child participants actually were, how factors like SES, educational level of parents, kind of school and teaching practices typically prevalent in the particular society were taken into account. This is obviously important as we know that in many parts of the world, rote learning as opposed to individual teaching prevails, including corporeal punishment which could also play a role in the observed variability and invariability across age and societies. These various factors need to be accounted for, with more details regarding the developmental niche of the participants, particularly in light of the great variability across cultures regarding the overall reliance on social information in general, increasing, decreasing or stable as a function of age, depending on the 7 societal extractions of the children. This instability needs to be better acknowledged and accounted for.

The significantly greater tendency across societies of males compared to females in using social information is intriguing and could be addressed in reference to other findings that might exist in the literature, in particular recent cross-cultural studies on the development of equity and pro-

social behaviors.

Was there any test of comprehension and retention of the task following the video presentation and prior to testing? Not clear as it stands and would be important as a control before inclusion in the data set. Could, for example, choices toward the non-demonstrated option (so-called Innovation), be predicted by a lack of comprehension and information retention of the child? In other words, what does it mean for a child to choose the third, non demonstrated tube? Why would a child choose the third option? The question is important, psychologically meaningful, and not specifically addressed, as far as I can tell. It should. What predicts the prevalence of such third choice across cultures? Is it linked to a depleted tendency to conform? Or to a proclivity toward "innovation" (thinking independently and outside of the box) as suggested by the authors. Not sure and worthwhile mentioning.

Reviewer #2 (Remarks to the Author):

General:

The authors take a cultural approach to investigating a complex question relating to social learning. Specifically, they examined children's reliance on social information and innovation as well as preference for majority versus minority influences.

They find that children's propensity to rely on social information is cultural variable whereas there is consistency across cultures in majority preference, yet the pattern is dependent on age (U-shaped function). The authors should be commended on their effort and ability to recruit and test such a wide age range across such diverse societies. The study, analyses, and results were conducted and communicated clearly. My suggestions and questions below are minor.

In my view the study makes a significant contribution to the field, but it is limited in the interpretation of the results as well as the predictive value moving forward.

- Figure 3 (a) If we examine the two extreme – opposing slopes – Indonesia and Brazil/CAR and, if I understand the figures correctly, increasing proportions (y-axis) correspond to the proportion of children "using social information" or selecting one of the two tubes that were demonstrated. If this is correct, how can we interpret the differences between Indonesian and Brazilian/CAR children? What might lead to such different approaches?
- Limitations – the limitations and alternative possible explanations for the behaviors observed must be recognized in the manuscript.
- Line 132 regarding "natural pedagogy" being culturally determined is strong, given that this is not a test of (and, it's a one-shot experiment, with little other information to support these findings) capacity to use social cues. In fact, if I read the natural pedagogy literature clearly, without the social cues, Gergely & Csibra would not have strong predictions regarding the child's tube choice. Were there social cues – IDS, direct eye gaze? Perhaps elaborating on this point or removing it from the manuscript would help.
- Line 143 – what is an "early reflexive majority bias"?

Additional questions/comments: Why a one-shot design? (needs justification)

- consider citing Broesch, Rochat and Itakura, 2017, where we examine children's model choice in 3 cultures (selective requests with age – we find cultural differences in the propensity to request)
- consider citing Legare's recent work (2017) on conformity bias.

Reviewer #3 (Remarks to the Author):

The authors present a cross-cultural study of social learning in young children. They use a task with 3 outcomes which can be categorized as the innovative option, majority option and minority option. The authors find that children's proclivity to innovate changes with age and varies cross culturally, but when they do use social information there is little difference between cultures wrt to choosing the majority or minority option and instead all exhibit a u-shaped pattern over development.

Overall, I think this is a valuable piece of work and the authors should be commended for carrying out such a large-scale study. As they mention in the introduction there is very little work on the cross-cultural variation in the development of social learning and so this is a valuable contribution. That said, I have a number of issues with the presentation of the work that need to be addressed before I can recommend publication and so, as it stands, I recommend major revision.

The largest issue with the manuscript is that the reader has to rely very heavily on the supplementary information to work out details of the analysis and results. I suspect that this manuscript was previously submitted to another journal with tighter space requirements (e.g. Nature) and so this is a legacy of that. This is fine, but Nature Communications allows authors much more room and so I think the authors should take advantage of this by moving much of the supplementary material into the main paper and expanding on various parts of the paper. Here are some specific suggestions:

lines 57-69 - the authors should describe at least one of these studies in more detail - what was done? what tasks were used? What were the results? etc.

lines 70-77 - same as above, these studies are so clearly relevant to the current work the reader needs a more detailed description.

lines 101-109 - trying to understand the results is very difficult without first reading section 3.2 of the SI. I would move as much of that material as possible into the main paper. In particular the description of the model (which effects were included, and so on) is critical. The authors may want to look into Richard McElreath's recent textbook "statistical rethinking" as I think it uses a way to describe the structure of models that is relatively straightforward and easy to understand.

lines 135-143 - the manuscript needs a more thorough discussion of u-shaped curves in developmental psychology. An example or two would be good, and a more thorough explanation of why they are a common phenomenon as well as the authors proposed explanation for this particular case.

discussion - in general the discussion is very light and needs beefing up. A handful of questions I was left with at the end but were not addressed are: (1) does it matter that all children watched a video of German children? We know that young children copy differentially according to cues of ethnicity so my gut is that yes, this probably does matter. The authors should discuss this. see: Kinzler, Katherine D, Kathleen H Corriveau, and Paul L. Harris. 2011. "Children's Selective Trust in Native-Accented Speakers." *Developmental Science* 14 (1): 106-11. doi:10.1111/j.1467-7687.2010.00965.x. (2) The authors findings actually different from one of the only other studies of the development of a majority bias: Morgan, Thomas J.H., Kevin N. Laland, and Paul L. Harris. 2014. "The Development of Adaptive Conformity in Young Children: Effects of Uncertainty and Consensus." *Developmental Science*, September. doi:10.1111/desc.12231. Briefly, while the authors find majority bias decreases between the ages of 3 and 7, the 2014 paper finds it increases. This should be discussed. (3) (some of) The authors have used a similar task previously: Haun, Daniel B M, Yvonne Rekers, and Michael Tomasello. 2012. "Majority-Biased Transmission in Chimpanzees and Human Children, but Not Orangutans." *Current Biology* 22 (8). Elsevier Ltd: 727-31. doi:10.1016/j.cub.2012.03.006. but the manuscript doesn't attempt to

compare the results of the two projects. I wonder if anything could be drawn from a comparison of the two?

Beyond this I have a few other points:

Main paper:

lines 47-56 - This paragraph is hard to follow, I suggest rewording it.

Figure 3 - the figure legend says Tanzania, but the manuscript says Kenya - which is it?

Figure 3b - the authors state there is no effect of culture wrt majority vs. minority, but the curves for the different cultures seem so different (at least as different as the lines in the previous panel) - what's going on here? Moreover, in the SI the authors mention that non-significant effects were removed from the model, but if culture has no significant effects on this panel then why do the lines look different at all?

lines 123-124 - the authors state that the demonstrators conferred adaptive information, but I don't really see what's adaptive about it.

lines 168-172 - the comparison here is inappropriate for 2 reasons: (1) different levels of significance for two groups against the same null hypothesis does not constitute evidence of a difference between the 2 groups, and (2) the authors seem to have compared the 3 lowest scoring groups against the 4 highest scoring groups. However, choosing which data to compare on the basis of how they perform is wildly inappropriate and should never be done. To illustrate, imagine the authors first grouped the data according to which day of the week the data were collected on. There will, undoubtedly, be variation between days and if the authors were to pool data from the 3 lowest scoring days and compare it to the 4 highest scoring days they may well conclude that day of the week is highly important. But this is simply because pooling the data according to the data itself is circular reasoning that will distort your results.

Supplementary information

Table S1 - it would be nice to have $n=...$ for each of the drop out criteria, some of them are quite vague (e.g. "subject seemed not to understand the instructions") and if they were used a lot it makes me suspicious of the results.

Table S2 - how can the two analyses end up with different n ? This seems like a big deal and I would like to see it discussed in detail in the main paper.

Section 3.1 - even with the authors additional tests opportunistic data collection will forever remain problematic for tests that yield p -values. That is because p -value calculations make assumptions about the amount of data the experimenter intended to collect (specifically they assume you collected precisely the amount of data you set out to collect). With opportunistic data collection this assumption is very aggressively violated and so we should treat all p -values with caution. There is very little the authors can do about this, however, in future I suggest they switch to using Bayesian stats which do not make such assumptions.

Section 3.2.1 - the authors used 2 binomial models, however a single categorical/multinomial model would have sufficed. Is there any reason for this choice?
Also, the authors used binomial models, but I understand that individual data points were not combined (i.e. it was a binomial model where $n=1$) - however, this is simply a Bernoulli model and so the authors may as well call it that instead.

Section 3.2.2b - the authors find a huge effect of order of demonstration on subsequent decision

making. This absolutely needs to be included in the main paper. The marginal effects of sex are also interesting as they point in the opposite direction to most adult studies (i.e. women are usually assumed to be more conformist than men, for a recent discussion of this see: Cross, Catharine P., Gillian R. Brown, Thomas J. H. Morgan, and Kevin N. Laland. 2016. "Sex Differences in Confidence Influence Patterns of Conformity." *British Journal of Psychology*, 1-13. doi:10.1111/bjop.12232.).

Response to Reviewers' comments:

Reviewer #1 (Remarks to the Author):

Review for Nature Communications

Manuscript #

NCOMMS-18-02428

Majority influence in human social learning develops universally across seven societies
Edwin J. C. van Leeuwen, Emma Cohen, Emma Collier-Baker, Christian J. Rapold, Marie Schäfer, Sebastian Schütte, Daniel B. M. Haun

This is an elegant and clever cross-cultural and developmental studies showing both variable and stable tendencies in children from 7 different societies aged between 4 and 14 years in their reliance on social information from peers in a 3 choice puzzle (3 possible tubes to drop one ball) in order to obtain a reward. With a sample of over 600 participants of various ages, not strictly matched in terms of age and numbers across 7 cultures that were, except for Germany (the reference, overloaded culture), opportunistically tested and with (as it stands) no other articulated rationale than being different and covering 4 continents.

Data are reported suggesting a) that across societies, children vary significantly in their overall tendency to rely on social information, and b) that across societies, there is a U shape developmental trend toward an early and late enhanced reliance on social information from a majority as opposed to a minority, with an invariant deflection or decrease of such tendency between 6 and 10 years of age.

These findings are certainly meaningful and stimulating, opening up new questions and demonstrating an economical way to study experimentally child development from a cross-cultural perspective, a long neglected domain. The paradigm is remarkably simple yielding rich information on the perennial issue of cultural factors in child development, in particular the variable and invariant aspects of the way children from different societies tend to rely on the larger (majority) group to learn and access information, the degree to which they are inclined to conform and learn from others in general, a traditional problem in both developmental and social psychology.

Our response: We thank Reviewer #1 for the positive evaluation of our manuscript.

The apparently trans-cultural U shape developmental progression in the tendency to rely on majority as opposed to minority information is intriguing and begs for more research on the phenomenon. This research is needed to provide more specifics regarding the “changing of social motivations” proposed by the authors. The report leaves open the question of what is exactly the psychology of the younger children relying more on the majority rather than the minority information? How different is it from the older children showing the same trend? What accounts for the drop in-between? If the phenomenon holds, it could revive and would open a whole new body of cross-cultural and developmental studies.

Our response: We have now elaborated on the possible explanation for the U-shaped developmental trajectory with respect to the majority bias (lines 166-196). We agree that this finding is highly relevant for the field and warrants systematic cross-cultural developmental

studies (see also our response to Reviewer #1's next point).

This said, the weakest point of the paper -as I see it- is the acknowledged opportunistic choice of societies (except for Germany), with no other rationale provided nor any information regarding who the child participants actually were, how factors like SES, educational level of parents, kind of school and teaching practices typically prevalent in the particular society were taken into account. This is obviously important as we know that in many parts of the world, rote learning as opposed to individual teaching prevails, including corporeal punishment which could also play a role in the observed variability and invariability across age and societies. These various factors need to be accounted for, with more details regarding the developmental niche of the participants, particularly in light of the great variability across cultures regarding the overall reliance on social information in general, increasing, decreasing or stable as a function of age, depending on the 7 societal extractions of the children. This instability needs to be better acknowledged and accounted for.

Our response: Thank you for raising this point. We have purported to showcase the necessity of taking into account a cultural perspective in the study of human ontogeny and human behavioural universals. Our findings show that regardless of the specific factors underlying the observed patterns (which, to be honest, will be difficult to measure/obtain retrospectively), a cultural approach seems key to understanding human ontogeny. This ground-breaking finding indeed warrants follow-up studies that aim at identifying the specific aspects of cultures that are linked to outcomes in the domain of cultural learning (lines 215-222).

The significantly greater tendency across societies of males compared to females in using social information is intriguing and could be addressed in reference to other findings that might exist in the literature, in particular recent cross-cultural studies on the development of equity and pro-social behaviors.

Our response: We have now moved our sex-specific findings from the Supplements to the main text and discuss our results in a broader (conformity) perspective in lines 197-209.

Was there any test of comprehension and retention of the task following the video presentation and prior to testing? Not clear as it stands and would be important as a control before inclusion in the data set. Could, for example, choices toward the non-demonstrated option (so-called Innovation), be predicted by a lack of comprehension and information retention of the child? In other words, what does it mean for a child to choose the third, non demonstrated tube? Why would a child choose the third option? The question is important, psychologically meaningful, and not specifically addressed, as far as I can tell. It should. What predicts the prevalence of such third choice across cultures? Is it linked to a depleted tendency to conform? Or to a proclivity toward "innovation" (thinking independently and outside of the box) as suggested by the authors. Not sure and worthwhile mentioning.

Our response: Thank you. We have meticulously ascertained that the test conditions were optimally valid by applying strict exclusion criteria, such that children who showed signs of a lack of understanding were not included in the data-analysis (see Table S1). We have now incorporated the following sentence in our manuscript: "Children's responses were only used if children's comprehension of the task was judged as valid by the experimenter onsite as well as two independent video-coders (see Table S1)." (lines 115-116).

Reviewer #2 (Remarks to the Author):

2018-02-12

General:

The authors take a cultural approach to investigating a complex question relating to social learning. Specifically, they examined children's reliance on social information and innovation as well as preference for majority versus minority influences.

They find that children's propensity to rely on social information is cultural variable whereas there is consistency across cultures in majority preference, yet the pattern is dependent on age (U-shaped function). The authors should be commended on their effort and ability to recruit and test such a wide age range across such diverse societies. The study, analyses, and results were conducted and communicated clearly. My suggestions and questions below are minor.

Our response: We thank Reviewer #2 for the positive evaluation of our manuscript.

In my view the study makes a significant contribution to the field, but it is limited in the interpretation of the results as well as the predictive value moving forward.

Our response: Thank you, we have now broadened our interpretation and predictive value, please see our responses to the specific points raised below.

• Figure 3 (a) If we examine the two extreme – opposing slopes – Indonesia and Brazil/CAR and, if I understand the figures correctly, increasing proportions (y-axis) correspond to the proportion of children “using social information” or selecting one of the two tubes that were demonstrated. If this is correct, how can we interpret the differences between Indonesian and Brazilian/CAR children? What might lead to such different approaches?

Our response: The observed cultural variability in the use of social information is striking and will instigate more specific studies investigating the underlying mechanisms. Here, we can only speculate, and thus highlight that “future research with a selected set of contrasting cultures might examine whether a relatively high socio-economic standard (Germany and Indonesia (Jakarta) in our sample) might buffer individuals against potential costs incurred by individual exploration¹ and consequently induce innovation.” (lines 157-160 & 215-222).

Specific to the example raised by Reviewer #2: The concerning Indonesian sample consisted of children seemingly living under better socio-economic conditions than the children living in the selected villages in Brazil and Central Republic Africa (see Supplementary Information). We speculate that societies with higher socio-economic standards may instigate more innovative behaviour in their residents for the reason that failure to make the correct decision (and thus potentially lose out on valuable resources) has less far-reaching consequences than in societies with low socio-economic standards (i.e., lacking a financial “safety-net”). This speculation is based on extensive research showing that individual learning (here: innovation) is more risk-prone than social learning (here: following minority/majority).

• Limitations – the limitations and alternative possible explanations for the behaviours observed must be recognized in the manuscript.

Our response: In general, we agree. For instance, we have now discussed the alternative explanation that children from different societies might respond differently to demonstrations from Caucasian/German children (see lines 223-231). At the same time, we counter this alternative explanation by pointing out that the cultural variability as observed in the social information use results are not consistent with a bias in which non-Caucasians prefer/avoid Caucasian models, and by referring to the robust U-shaped developmental pattern with respect to the majority bias, which cannot be explained by an ethnic bias, because both the majority and minority consisted of Caucasian children.

• Line 132 regarding “natural pedagogy” being culturally determined is strong, given that this is not a test of (and, it’s a one-shot experiment, with little other information to support these findings) capacity to use social cues. In fact, if I read the natural pedagogy literature clearly, without the social cues, Gergely & Csibra would not have strong predictions regarding the child’s tube choice. Were there social cues – IDS, direct eye gaze? Perhaps elaborating on this point or removing it from the manuscript would help.

Our response: Thank you for pointing this out. We have aimed to contrast natural versus cultural pedagogy for the reason that our findings seem to indicate that cultural practices (e.g., pedagogy) can affect social learning biases, which would speak in favour of cultural pedagogy. However, we agree that we do not report any “pedagogy” specifics, and so we have removed this point from the manuscript (see lines 160-163).

• Line 143 – what is an “early reflexive majority bias”?

Our response: We have now clarified this in the manuscript: “An early reflexive majority bias refers to the proposition that young children may irrationally follow the majority for reasons of finding safety in social proximity. In later life, this reflexive majority bias may come to develop into a rational decision reflecting individuals’ aim to make correct decisions (i.e., informational conformity) or avoid social repercussions for being different (i.e., social conformity).” (lines 182-187).

Additional questions/comments: Why a one-shot design? (needs justification)

Our response: We elaborate on this decision in the Supplementary Information. We viewed the one-shot design as a stronger means to obtain data independency than statistically correcting for repeated measures (also see line 259).

- consider citing Broesch, Rochat and Itakura, 2017, where we examine children’s model choice in 3 cultures (selective requests with age – we find cultural differences in the propensity to request)**
- consider citing Legare’s recent work (2017) on conformity bias.**

Our response: Thank you for these useful literature suggestions, we have now incorporated this work into the manuscript (lines 73-76, and 237).

Reviewer #3 (Remarks to the Author):

The authors present a cross-cultural study of social learning in young children. They use a task with 3 outcomes which can be categorized as the innovative option, majority option and minority option. The authors find that children's proclivity to innovate changes with age and varies cross culturally, but when they do use social information there is little difference between cultures wrt to choosing the majority or minority option and instead all exhibit a u-shaped pattern over development.

Overall, I think this is a valuable piece of work and the authors should be commended for carrying out such a large-scale study. As they mention in the introduction there is very little work on the cross-cultural variation in the development of social learning and so this is a valuable contribution. That said, I have a number of issues with the presentation of the work that need to be addressed before I can recommend publication and so, as it stands, I recommend major revision.

Our response: We thank Reviewer #3 for the positive evaluation of our manuscript.

The largest issue with the manuscript is that the reader has to rely very heavily on the supplementary information to work out details of the analysis and results. I suspect that this manuscript was previously submitted to another journal with tighter space requirements (e.g. Nature) and so this is a legacy of that. This is fine, but Nature Communications allows authors much more room and so I think the authors should take advantage of this by moving much of the supplementary material into the main paper and expanding on various parts of the paper.

Our response: Thank you for pointing this out. We agree and have now moved all relevant information concerning data analysis and results from the Supplementary Information to the main manuscript (lines 115-149).

Here are some specific suggestions:

lines 57-69 - the authors should describe at least one of these studies in more detail - what was done? what tasks were used? What were the results? etc.

Our response: This is a very helpful suggestion. We have now described two studies (including their results) in this paragraph in more detail (lines 67-76).

lines 70-77 - same as above, these studies are so clearly relevant to the current work the reader needs a more detailed description.

Our response: Thank you. We have now described and interpreted the referenced studies in detail such that their relevance to our study becomes immediately clear (lines 91-97).

lines 101-109 - trying to understand the results is very difficult without first reading section 3.2 of the SI. I would move as much of that material as possible into the main paper. In particular the description of the model (which effects were included, and so on) is critical. The authors may want to look into Richard McElreath's recent textbook "statistical rethinking" as I think it uses a way to describe the structure of models that is relatively straightforward and easy to understand.

Our response: We have now moved all relevant parts concerning the data-analysis to the main manuscript, including the model specifications. In line with Reviewer #3's suggestion, we have followed McElreath's textbook structure in presenting the model specifications (lines 115-131).

lines 135-143 - the manuscript needs a more thorough discussion of u-shaped curves in developmental psychology. An example or two would be good, and a more thorough explanation of why they are a common phenomenon as well as the authors proposed explanation for this particular case.

Our response: Thank you. We have now provided an illustrative example of how U-shaped developmental curves may come about (lines 167-175), and how the U-shape curve might be explained for our results on the developmental trajectory of the majority bias (lines 175-196).

discussion - in general the discussion is very light and needs beefing up. A handful of questions I was left with at the end but were not addressed are: (1) does it matter that all children watched a video of German children? We know that young children copy differentially according to cues of ethnicity so my gut is that yes, this probably does matter. The authors should discuss this. see: Kinzler, Katherine D, Kathleen H Corriveau, and Paul L. Harris. 2011. "Children's Selective Trust in Native-Accented Speakers." *Developmental Science* 14 (1): 106-11. doi:10.1111/j.1467-7687.2010.00965.x.

Our response: Thank you. We have now elaborated the discussion of our findings in line with Reviewer #3's suggestions. Specifically, we now discuss the alternative explanation that children from different societies might respond differently to demonstrations from Caucasian/German children (lines 223-231). Moreover, we discuss our sex-differences results and relate them to previous investigation of sex-specific conformist responses (lines 197-209).

(2) The authors findings actually different from one of the only other studies of the development of a majority bias: Morgan, Thomas J.H., Kevin N. Laland, and Paul L. Harris. 2014. "The Development of Adaptive Conformity in Young Children: Effects of Uncertainty and Consensus." *Developmental Science*, September. doi:10.1111/desc.12231. Briefly, while the authors find majority bias decreases between the ages of 3 and 7, the 2014 paper finds it increases. This should be discussed.

Our response: Thank you for this helpful suggestion. We have now incorporated a specific reference to Morgan et al.'s findings in lines 187-190. We note, however, that our two studies are incompatible on the level of subtle study designs and conceptualizations to the point that discussion in our paper would side-track from the main (culture-centred) message. For instance, Morgan et al.'s focus is on varying majority sizes (while this is kept constant in our study), and they investigate subjects' tendency to change their initial decision, while we have sampled truly naïve children, without any indications of the truth-value of any solution.

(3) (some of) The authors have used a similar task previously: Haun, Daniel B M, Yvonne Rekers, and Michael Tomasello. 2012. "Majority-Biased Transmission in Chimpanzees and Human Children, but Not Orangutans." *Current Biology* 22 (8). Elsevier Ltd: 727-31. doi:10.1016/j.cub.2012.03.006. but the manuscript doesn't attempt to compare the results of the two projects. I wonder if anything could be drawn from a comparison of the two?

Our response: Thank you for pointing this out. We have now referred to the main message conveyed by Haun et al. (2012): the majority bias is (partly) shared with the other great apes, and thus highlights the fundamental and adaptive nature of this learning bias, see lines 71-73, and 215 (added reference).

Beyond this I have a few other points:

Main paper:

lines 47-56 - This paragraph is hard to follow, I suggest rewording it.

Our response: Thank you. We have now re-written this paragraph and describe the characteristics of individual and social learning, including their trade-off, more clearly (lines 46-61).

Figure 3 - the figure legend says Tanzania, but the manuscript says Kenya - which is it?

Our response: Our apologies, we have now corrected Figure 3 to say “Kenya”.

Figure 3b - the authors state there is no effect of culture wrt majority vs. minority, but the curves for the different cultures seem so different (at least as different as the lines in the previous panel) - what's going on here? Moreover, in the SI the authors mention that non-significant effects were removed from the model, but if culture has no significant effects on this panel then why do the lines look different at all?

Our response: Thank you for pointing this out. Most of the variation is explained by the U-shaped developmental curve, which is represented by the squared age term in the model. With respect to omitting the non-significant terms from the model, we referred to the “higher-order parameters (e.g., the interaction and squared terms)”, not the individual parameters (lines 284-286).

lines 123-124 - the authors state that the demonstrators conferred adaptive information, but I don't really see what's adaptive about it.

Our response: We referred to the fact that the demonstrators only show behaviour that reliably leads to the children obtaining the resource at stake, i.e., obtaining the reward is contingent on entering the ball into a (specific) hole of the apparatus. We have now clarified this in the manuscript (lines 152-153).

lines 168-172 - the comparison here is inappropriate for 2 reasons: (1) different levels of significance for two groups against the same null hypothesis does not constitute evidence of a difference between the 2 groups, and (2) the authors seem to have compared the 3 lowest scoring groups against the 4 highest scoring groups. However, choosing which data to compare on the basis of how they perform is wildly inappropriate and should never be done. To illustrate, imagine the authors first grouped the data according to which day of the week the data were collected on. There will, undoubtedly, be variation between days and if the authors were to pool data from the 3 lowest scoring days and compare it to the 4 highest scoring days they may well conclude that day of the week is highly important. But this is simply because pooling the data according to the data itself is circular reasoning that will distort your results.

Our response: We agree with Reviewer #3. We had aimed to convey the message that if the study had only focused on a narrow age range in two or three cultures with strongly different features, the results would have looked drastically different. Thank you for pointing out that the illustrative analysis in our manuscript is not correct. We have now omitted it from the manuscript and instead made the respective point by describing the issues at stake (lines 232-239).

Supplementary information

Table S1 - it would be nice to have $n=...$ for each of the drop out criteria, some of them are quite vague (e.g. "subject seemed not to understand the instructions") and if they were used a lot it makes me suspicious of the results.

Our response: We have now provided the n 's for the drop-outs (see Table S1). The assessment of the drop-outs was done independently by three experimenters. Regarding the n 's, maximally, there are 3 drop-outs for one *a priori* formulated criterion. There is only one drop-out for the criterion "subject seemed not to understand the instructions", and this coincided with criterion "Subject was not able to access all holes of the box equally".

Table S2 - how can the two analyses end up with different n ? This seems like a big deal and I would like to see it discussed in detail in the main paper.

Our response: We have discussed in the main paper that we have formulated two specific predictions based on existing literature (1. social information use versus individual exploration, and 2. Majority versus minority preference), and that we have subjected our data as accurately as possible to these predictions. This means that first, we have tested children's differential responses for social (any demonstration, thus both majority and minority children) versus individual learning (the non-demonstrated option), and second, children's specific preference for the majority (i.e., the majority children) versus the minority (the minority child). Hence, the second analysis leaves out the non-demonstrated option, or, in other words, consists of the subset of children who chose the social over the individual option (lines 115-131).

Section 3.1 - even with the authors additional tests opportunistic data collection will forever remain problematic for tests that yield p-values. That is because p-value calculations make assumptions about the amount of data the experimenter intended to collect (specifically they assume you collected precisely the amount of data you set out to collect). With opportunistic data collection this assumption is very aggressively violated and so we should treat all p-values with caution. There is very little the authors can do about this, however, in future I suggest they switch to using Bayesian stats which do not make such assumptions.

Our response: We thank Reviewer #3 for this suggestion and take it to heart.

Section 3.2.1 - the authors used 2 binomial models, however a single categorical/multinomial model would have sufficed. Is there any reason for this choice? Also, the authors used binomial models, but I understand that individual data points were not combined (i.e. it was a binomial model where $n=1$) - however, this is simply a Bernoulli model and so the authors may as well call it that instead.

Our response: The reason for using two binomial models instead of a single multinomial model lies in the conceptualization of our research questions. As mentioned before, we had derived two predictions from the literature, both of which required slightly different analysis. The first model assesses general reliance on social information (versus individual exploration), while the second model assesses within the sub-sample of children who choose the social over the individual whether the majority is preferred over the minority. We have now made this approach more explicit in the manuscript (lines 115-132). Moreover, we have now restricted our use of “binomial” to statements of the error distribution we used to model our data.

Section 3.2.2b - the authors find a huge effect of order of demonstration on subsequent decision making. This absolutely needs to be included in the main paper. The marginal effects of sex are also interesting as they point in the opposite direction to most adult studies (i.e. women are usually assumed to be more conformist than men, for a recent discussion of this see: Cross, Catharine P., Gillian R. Brown, Thomas J. H. Morgan, and Kevin N. Laland. 2016. "Sex Differences in Confidence Influence Patterns of Conformity." *British Journal of Psychology*, 1-13. doi:10.1111/bjop.12232.).

Our response: Thank you for pointing this out. We have now moved both our results with respect to order and sex effects to the main paper and interpret the order effect in terms of the mentioned primacy phenomenon (lines 144-149) and the sex effects in light of existing literature on sex differences with respect to conformist responses, including the suggested study by Cross et al. 2016 (line 197-209).

REVIEWERS' COMMENTS:

Reviewer #1 (Remarks to the Author):

I think the authors responded appropriately and thoroughly to all my and other reviewers' comments. I recommend publication. An important contribution to the field, opening new research venues on child development taking culture seriously.

Reviewer #2 (Remarks to the Author):

Dear Authors,

Thank-you for attending carefully to the details of my review. I feel that you have addressed all of my questions and concerns in the manuscript. I look forward to seeing the final product in print. All the best!

Tanya Broesch

Reviewer #3 (Remarks to the Author):

The authors have done an excellent job of responding to my previous comments and I think the manuscript is now suitable for publication. With this said, if the editor is feeling picky I have two remaining points that the authors may want to address:

One specific point: line 140 - the authors should not take $p > 0.05$ as evidence for cultural homogeneity, rather it is a lack of evidence for cultural heterogeneity. Minor detail, but technically important.

One general point: I'd be cautious about describing the majority preference as "intractable" - the data show it is highly culturally variable and often dips below 0.5, and while its usually U-shaped, this is not the case for Kenya, or Indonesia (see Figure 3B). To me this suggests it is just as malleable as social information use (Figure 3A). I worry that readers may come away with the message that "social information use is culturally malleable but majority bias is not" which doesn't really do the data justice: both social information use and majority preference show clear signs of cultural malleability with some shared (or "universal") patterns across cultures too, such as a U-shaped pattern for majority preference. I feel this is a more nuanced and appropriate general message.

Response to Reviewers' comments:

Reviewer #1 (Remarks to the Author):

I think the authors responded appropriately and thoroughly to all my and other reviewers' comments. I recommend publication. An important contribution to the field, opening new research venues on child development taking culture seriously.

Reviewer #2 (Remarks to the Author):

Dear Authors,

Thank-you for attending carefully to the details of my review. I feel that you have addressed all of my questions and concerns in the manuscript. I look forward to seeing the final product in print. All the best!

Tanya Broesch

Our response: We thank Reviewers #1 & #2 for the positive evaluation.

Reviewer #3 (Remarks to the Author):

The authors have done an excellent job of responding to my previous comments and I think the manuscript is now suitable for publication. With this said, if the editor is feeling picky I have two remaining points that the authors may want to address:

One specific point: line 140 - the authors should not take $p > 0.05$ as evidence for cultural homogeneity, rather it is a lack of evidence for cultural heterogeneity. Minor detail, but technically important.

Our response: We agree with Reviewer #3 and changed the text accordingly as follows:
"Comparing cultures, there was no evidence for cross-cultural heterogeneity in the U-shaped ontogenetic pattern for the majority preference (LRT culture/age² interaction: $\chi^2 = 3.98$, $df = 6$, $p = 0.679$; Figure 3b and Supplementary Fig. 2)." (Lines 146-149)

One general point: I'd be cautious about describing the majority preference as "intractable" - the data show it is highly culturally variable and often dips below 0.5, and while its usually U-shaped, this is not the case for Kenya, or Indonesia (see Figure 3B). To me this suggests it is just as malleable as social information use (Figure 3A). I worry that readers may come away with the message that "social information use is culturally malleable but majority bias is not" which doesn't really do the data justice: both social information use and majority preference show clear signs of cultural malleability with some shared (or "universal") patterns across cultures too, such as a U-shaped pattern for majority preference. I feel this is a more nuanced and appropriate general message.

Our response: Again, we agree with Reviewer #3 and changed the title, abstract (Lines 33-35) and Discussion (Lines 345-346) accordingly.